



# Large Herbivores Affecting Permafrost - Impacts of Grazing on Permafrost Soil Carbon Storage in Northeastern Siberia

Torben Windirsch[1,2], Guido Grosse[1,2], Mathias Ulrich[3], Bruce C. Forbes[4], Mathias Göckede[5], Juliane Wolter[1,6], Marc Macias-Fauria[7], Johan Olofsson[8], Nikita Zimov[9], and Jens Strauss[1]

[1]Alfred Wegener Institute Helmholtz Centre for Polar and Marine Research, Potsdam, Germany
[2]University of Potsdam, Department for Geosciences, Potsdam, Germany
[3]University of Leipzig, Institute for Geography, Leipzig, Germany
[4]Arctic Centre, University of Lapland, Rovaniemi, Finland
10   [5]Max Planck Institute for Biogeochemistry, Department Biogeochemical Signals, Jena, Germany
[6]University of Potsdam, Institute of Biochemistry and Biology, Potsdam, Germany
[7]University of Oxford, School of Geography and the Environment, Biogeosciences Lab, Oxford, Great Britain
[8]Umeå University, Department of Ecology and Environmental Science, Umeå, Sweden
[9]Northeast Science Station, Pacific Institute for Geography, Far-Eastern Branch of Russian Academy of Science, Chersky,
15   Republic of Sakha (Yakutia), Russia

*Correspondence to:* Torben Windirsch (torben.windirsch@awi.de)

**Key words:** organic material inventory, animal husbandry, bio-geo interactions, rewilding, climate change



**Abstract.** The risk of carbon emissions from permafrost ground is linked to ground temperature and thus in particular to thermal insulation by vegetation and organic soil layers in summer and snow cover in winter. This ground insulation is strongly influenced by the presence of large herbivorous animals browsing for food. In this study, we examine the potential impact of large herbivore presence on the ground carbon storage in thermokarst landscapes of northeastern Siberia. Our aim is to understand how intensive animal grazing may affect permafrost thaw and hence organic matter decomposition, leading to different ground carbon storage, which is significant in the active layer. Therefore, we analysed sites with differing large herbivore grazing intensity in the Pleistocene Park near Chersky and measured maximum thaw depth, total organic carbon content and decomposition state by $\delta^{13}C$ isotope analysis. In addition, we determined sediment grain size composition as well as ice and water content. We found the thaw depth to be shallower and carbon storage to be higher in intensively grazed areas compared to extensively and non-grazed sites in the same thermokarst basin. The intensive grazing presumably leads to a more stable thermal ground regime and thus to increased carbon storage in the thermokarst deposits and active layer. However, the high carbon content found within the upper 20 cm on intensively grazed sites could also indicate higher carbon input rather than reduced decomposition, which requires further studies. We connect our findings to more animal trampling in winter, which causes snow disturbance and cooler winter ground temperatures during the average annual 225 days below freezing. This winter cooling overcompensates ground warming due to the lower insulation associated with shorter heavily grazed vegetation during the average annual 140 thaw days. We conclude that intensive grazing influences the carbon storage capacities of permafrost areas and hence might be an actively manageable instrument to reduce net carbon emission from these sites.

## 1 Introduction

In the context of global climate warming, carbon emissions from Arctic permafrost regions have been identified as a key source of greenhouse gases (GHG), further accelerating the permafrost carbon-climate feedback and increasing atmosphere warming (Schuur et al., 2015; Turetsky et al., 2019; Bowen et al., 2020). An estimated 1300 Petagrams (Pg) of carbon are stored within the upper 3 m of ground in the permafrost region, of which approximately 1000 Pg are perennially frozen (Hugelius et al., 2014; Mishra et al., 2021). The carbon storage and release mechanisms of permafrost had an important role for atmospheric GHG levels during the Late Quaternary (Zimov et al., 2006; Walter et al., 2007; Lindgren et al., 2018). With further Arctic warming now and in the future, permafrost thaws and organic material (OM), formerly frozen for centuries to tens of millennia, is expected to become widely available for microbial decomposition and increased GHG production (Schuur et al., 2008).

Vegetation composition and snow conditions play a major role in the Arctic carbon cycle and on the land surface energy budget. Plant growth itself affects above and below ground carbon storage as plants take up carbon dioxide from the environment to form new tissue, temporarily fixing carbon in their biomass, while plant litter and roots become part of the active layer soils. Overall plant biomass is higher in heath and tundra vegetation, compared to grasslands (Arndal et al., 2009). While a dense vegetation cover may reduce summer energy exchange between the ground and the atmosphere by creating wind protection and a stable air layer between the ground's surface and the canopy (Zhang et al., 2013; Mod and Luoto, 2016; te Beest et al., 2016), shrub vegetation also effectively traps snow in winter, leading to locally higher snow accumulation (Domine et al., 2016). This snow again insulates the ground underneath which prevents penetration of cold winter air temperatures and thus, in contrast to ground shading by shrubs in the summer, contributes to an increase of mean annual ground temperature (Sannel, 2020). In contrast, grasses facilitate ground cooling in winter, as they bend beneath accumulating snow, reducing the volume of air trapped beneath the snow, in comparison to shrub vegetation (Blok et al., 2010). In this way, grasses contribute to cooling the ground year-round, which might reduce permafrost thaw and hence prevents frozen OM from



degradation (Macias-Fauria et al., 2020). However, due to the warming climate and an associated extension of the growing seasons, an increase in the establishment and growth of shrub vegetation is already observed (Frost et al., 2013) and also projected to increase in the Arctic tundra regions (Zhang et al., 2013).

Some researchers have proposed to actively exploit these processes and properties affecting the surface energy balance in Arctic regions to preserve permafrost and to limit or reduce permafrost carbon emissions due to a warming climate. In particular, Zimov et al. (1995) proposed a return to the now extinct mammoth steppe biome, widespread in the terrestrial Arctic during the Last Glacial Maximum (about 21 kyr ago). The establishment of sufficiently large numbers of herbivores as ecosystem engineers could intensify grazing and trampling pressure in contemporary tundra and forest-tundra landscapes (Beer et al., 2020). During the late Pleistocene, the mammoth steppe was characterized by highly productive grasslands with high grazing pressure by large herbivores. Olofsson et al. (2004) and Zimov (2005) have suggested that grazing in the Arctic reduces shrub abundance. This could help shift shrubifying tundra ecosystems towards grasslands similar to the mammoth steppe in terms of productivity and thermal insulation properties. Shifting vegetation towards such grasslands is also suggested to cause a reduction in albedo and hence latent and sensible heat fluxes due to overall reduced energy input (te Beest et al., 2016). In winter, dense populations of large herbivores may also affect the insulation effect of snow on the ground by trampling down or removing the dense snow cover in search for forage, leading to enhanced refreezing of the ground (Beer et al., 2020).

Other studies have reported on the effect of herbivore presence - mainly reindeer - on carbon storage in tundra biomes (Olofsson, 2006; Falk et al., 2015; Olofsson and Post, 2018; Ylänne et al., 2018). Modelling studies have tried to quantify the impact of large herbivores on permafrost soil carbon storage (Zimov et al., 2009; Beer et al., 2020). These modelling exercises and predictions have however not yet been tested in the field. The Pleistocene Park project, near Chersky in northeastern Siberia, aims to re-establish a megaherbivore guild via rewilding of the tundra with greater numbers of modern large and cold-adapted herbivores such as musk oxen, Yakutian horses, Kalmyk cattle, bison and reindeer, among others. With the present study focusing on research sites in the Pleistocene Park, we aim to assess the influence of dense and functionally diverse herbivory pressure on soil carbon storage in terrestrial ecosystems in an ice-rich Arctic permafrost landscape. We hypothesize that a more intense large herbivore density and hence intensive grazing and trampling leads to increased carbon storage and reduced decomposition of the preserved OM.

In this study we compare belowground carbon and sediment characteristics between sites in two landscape units and with different grazing intensities. This was addressed with a multiproxy approach.

## 2 Study area

Our study area, the Pleistocene Park, is located in northeastern Siberia in the floodplains of the Kolyma River and approximately 100 km inland from the Arctic Ocean (Fig. 1)(see Fuchs et al. (2021)). The landscape is characterized by





thermokarst lakes and drained thermokarst basins of different depths and interspersed Yedoma uplands (Schirrmeister et al., 2011; Palmtag et al., 2015).

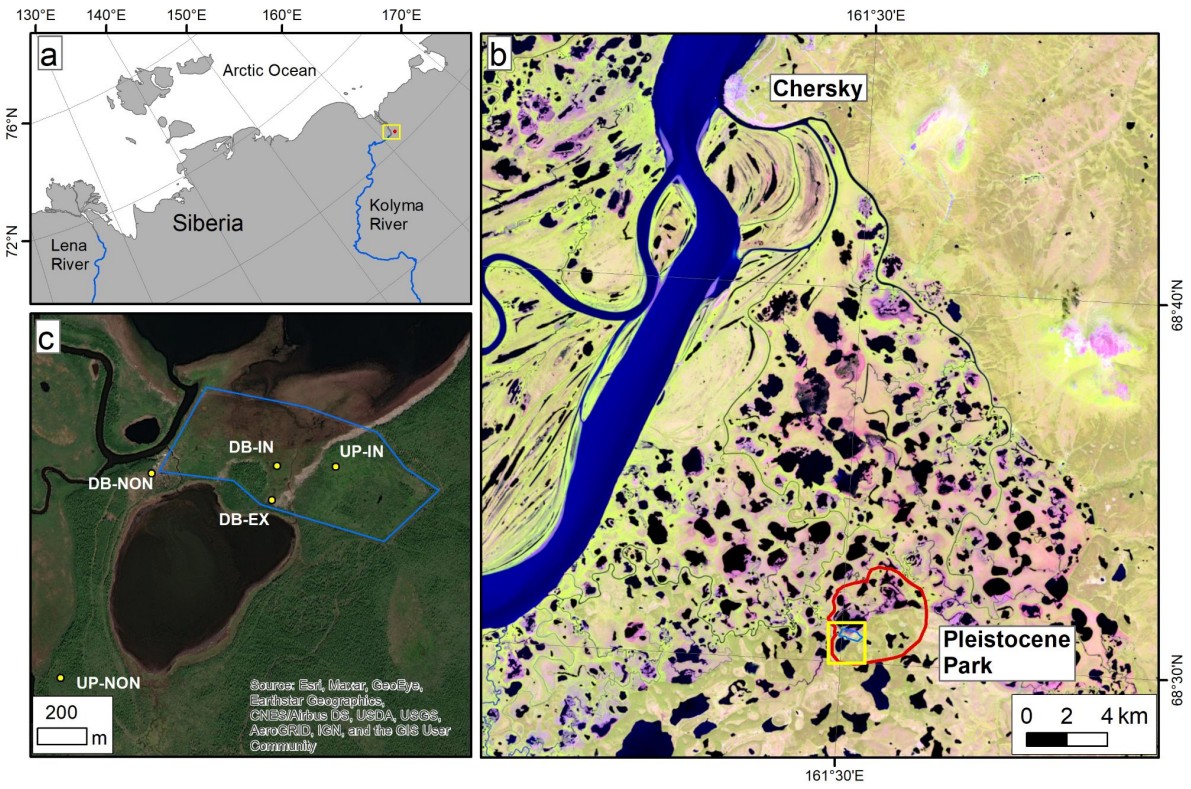

Figure 1 – Map of the study area location in northeastern Siberia marked with a yellow frame (a). In the Landsat-8
satellite image (band combination of shortwave infrared 1 - near infrared - red, 2019-07-04) of panel (b) the red line
marks the area of the Pleistocene Park south of Chersky; the yellow frame indicates the location of the field study sites
shown in (c). Panel (c) shows a high resolution WorldView satellite image (2018-07-21; provided by ESRI) with the
labeled sampling sites, with DB indicating a drained basin site; UP indicating an Yedoma upland site; IN (intensive),
EX (extensive), and NON (non-grazed) indicate the grazing intensity of each site.

Within the thermokarst basins, vegetation varies with the local wetness gradient and to some extent with grazing intensity.
*Carex appendiculata* tussocks dominate in shallow water around lake margins. In frequently flooded areas, tall grasses (e.g.
*Calamagrostis langsdorfii*) grow up to 70 cm high, which is a little higher than in surrounding areas (Corradi et al., 2005).
Seasonal flooding typically occurs after snow melt, temporarily refilling the drained thermokarst basins and covering study
site DB-IN (Fig. 1). Towards the outer basin margins, a driftwood belt was found (between DB-IN and DB-EX in our study
area). Beyond this driftwood belt on the elevated reaches of the basin, there is an extensively grazed grassland with occasional
forbs (DB-EX). Towards the edge of the basin, drier ground is covered with dense trees and shrubs (*Larix* sp., *Salix* sp.) of up





to 2.5 m height, which grow into forests in some places (Fig. 1, west of DB-IN and DB-EX). A mix of dwarf and taller shrubs (e.g. *Salix* sp.), forbs and grasses is found on the Yedoma uplands, where sites UP-IN and UP-NON are located. On the edge between basin and upland, the non-grazed site DB-NON is located just within the thermokarst basin.

In this landscape, the Pleistocene Park project was started as a large-scale long-term ecosystem change experiment in 1996 (Zimov, 2020). For this experiment, a large number of Yakutian horses, Kalmykian cattle, reindeer, bisons, musk oxen as well as moose, sheep, yaks and European bison – all large and cold-adapted herbivores – were gradually introduced to a 40 ha fenced area stretching across tundra and forest-tundra vegetation on Yedoma uplands and within partially drained thermokarst basins. Today, this 40 ha area is the heartland of a 160 km$^2$ fenced park with several experimental sites examining rewilding

impacts on Arctic ecosystems (Macias-Fauria et al., 2020). The presence of these animals - and some man-made interventions such as removal of trees to build fences around the park - has already transformed most of the previous tundra vegetation into grassland, while forest areas are developing towards more open vegetation. Long-term studies on greenhouse gas (GHG) emissions have been conducted in nearby areas of the Pleistocene Park (Göckede et al., 2017): on a nearby floodplain of the Kolyma River, Göckede et al. (2017); (2019) investigated the carbon and energy budgets of tundra wetlands and shifts thereof

related to drainage disturbance. They found that an undisturbed wetland, similar to the non-grazed wetlands in the Pleistocene Park area, acted as a moderate annual sink for $CO_2$, and as a moderate source for $CH_4$. Both carbon and energy cycles were shown to be highly sensitive to shifts in hydrology.

Climate in this region is characterized by large temperature amplitudes (average of -33 °C in January; average of 12 °C in July) with a mean annual temperature of -11 °C (Göckede et al., 2017). Annual precipitation sums up to 197 mm, with March

being the driest month (7 mm) and August having the most precipitation (30 mm). Main seasons for precipitation are summer and autumn (Göckede et al., 2017).

Prevailing deposit types are Yedoma (Strauss et al., 2017) and thermokarst deposits (Veremeeva et al., 2021) with the latter covering approximately 58 % of the land area in regions with high Yedoma coverage, and up to 96.4 % in regions with low Yedoma deposit occurrence. These deposits are intermitted by marshes, river valleys and deltas (Veremeeva et al., 2021).

## 130  3 Methods

### 3.1 Field sampling approach

The sampling sites were chosen based on their grazing intensity, which was identified by Pleistocene Park staff according to monitored animal preferences in grazing sites at the start of the field campaign in July 2019. Four sites were selected to cover different grazing intensities (intensive (IN), extensive (EX) and non-grazed (NON)) and landforms, including drained

thermokarst basin (DB) and surrounding Yedoma uplands (UP). The sites were sampled during our field campaign (Fig. 1): DB-IN (68.512694° N, 161.50875° E) as an intensively grazed site in the still-wet area of a thermokarst basin; DB-EX (68.511111° N, 161.508528° E) as an extensively grazed site within the thermokarst basin, close to the fence of Pleistocene Park; DB-NON (68.512167° N, 161.496278° E) as an ungrazed site within the thermokarst basin, just outside the park's fence;



UP-IN (68.512778° N, 161.514611° E) as an intensively grazed site on a Yedoma upland surrounding the DB thermokarst
basin. We chose the sites within the same landscape unit (for all DB sites and all UP sites) to have a similar basis in which the
animal influence is the main changing aspect and differences between all other characteristics are minimised. Since animal
trampling as well as defecation always occur along actual grazing, we will use the term "grazing" as a description for all animal
activity including trampling, defecation and foraging in the following.

Firstly, a description of the surroundings, including the main vegetation type was done at each sampling site.

Secondly, we removed the thawed layer using a spade, until we hit frozen ground. The excavated soil profile was sampled
using fixed-volume steel cylinders with a volume of 250 cm³. Due to very wet ground conditions, we were not able to collect
samples with a known volume at all sites. These wet soil parts occurred specifically in the intensively grazed thermokarst basin
site DB-IN and reached from the surface to the frozen ground (38 cm). Here, we cut blocks of 8 to 10 cm in height out of the
profile wall using a knife. The organic top layer was sampled separately.

Thirdly, for sampling the frozen ground underneath, we used a SIPRE permafrost auger with a diameter of 7.6 cm in order to
sample both still-frozen parts of the active layer and the underlying permafrost. We reached maximum sampling depths of 110
cm below surface (bs) at DB-IN, 108 cm bs at DB-EX, 127 cm bs at DB-NON and 114 cm bs at UP-IN (Fig. S2 and S3). After
drilling, soil samples and cores were individually wrapped in sterile plastic bags. All samples were brought in frozen state to
our labs for further analysis.

Due to high and dense shrub vegetation and therefore inaccessibility, we were not able to sample the location of UP-NON
(68.504469° N, 161.488390° E) during our summer field campaign. The core from this site was retrieved using the same
SIPRE auger but was drilled in winter in completely frozen soil, hence it's smaller length, reaching 72 cm bs. It was retrieved
from a Yedoma upland outside the fence of Pleistocene Park.

### 3.2 Laboratory work

The frozen core pieces were subsampled in the lab approximately every 5 cm according to stratigraphy using a bandsaw.
Afterwards, all samples were freeze-dried and weighed for ice and water content. The dry samples were split into subsamples
for biogeochemical and sedimentological analysis.

The samples for biogeochemical analysis were homogenized using a planetary mill (Fritsch Pulverisette 5) and weighed into
tin capsules and steel crucibles for measurement. We determined total carbon (TC), total nitrogen (TN) and total organic carbon
(TOC) by combustion analysis using an Elementar vario EL III and an Elementar soliTOC cube. Afterwards, the carbon-
nitrogen ratio (C/N) was calculated from TOC and TN, giving information about the state of degradation and the source of the
OM. C/N ratios could not be calculated for samples with TN or TOC below detection limit of 0.1 wt%.

OM for radiocarbon dating was taken from the dried original samples. For dating, the Mini Carbon Dating System (MICADAS,
see Gentz et al. (2017)) at the Alfred Wegener Institute Bremerhaven was used. We calculated the results in calibrated years
before present (cal yr BP) using the calibration software Calib 8.2 and applying the IntCal20 calibration curve (Reimer et al.,
2020; Stuiver et al., 2021).





We analyzed the ratio of stable carbon isotopes and used it as a proxy for the degree of decomposition of the OM following Diochon and Kellman (2008). Samples for $\delta^{13}C$ analysis were homogenized using a planetary mill, and subsequently treated with hydrochloric acid at 50 °C to remove carbonates. Measurement was done using a Delta V Advantage Isotope Ratio MS
supplement equipped with a Flash 2000 Organic Elemental Analyzer. Results are given in ‰ compared to the Vienna Pee Dee Belemnite (VPDB) standard (Coplen et al., 2006).

To determine grain size distribution within samples, OM was removed from the samples using hydrogen peroxide. The grain sizes were subsequently measured using a Malvern MasterSizer 3000.

### 3.3 Statistics and external data

We calculated the number of freezing days (daily mean air temperature below 0 °C) by calculating the average number of such days per year over nine years. For this we used data from 2009 to 2017 measured in Chersky (station RSM00025123). Data was obtained from the NOAA NCDC database.

Checking for significant differences between intensively grazed and non-grazed sites, we used a Mann-Whitney-U test with a confidence interval of 95 % on our TOC data combined for both intensive (DB-IN and UP-IN) and both non-grazed (DB-NON
and UP-NON) sites. This test was used as we checked for TOC variations in different sample sizes (intensive: n = 36 samples out of 5 cores; non-grazed: n = 29 samples). In addition we used the same test approach on combined intensive and non-grazed site samples for the uppermost 38 cm (intensive: n = 11 samples, non-grazed: n = 10 samples), which is the shallowest thaw depth we encountered and therefore part of the active layer at all sites. Since grazing intensities were artificially altered within the last decades, we expect the most pronounced differences in the seasonally thawed layer, which corresponds to more recent
periods. We visualized the statistical distribution of TOC contents in boxplots for intensive, extensive and non-grazed sites of our study, both for the entire core lengths and for the uppermost 38 cm to explore possible differences in TOC content in relation to grazing intensity.

To identify correlation between TOC content, water (respectively ice) content and sediment type (via mean grain size), we used principal component analysis (PCA). Data were initially normalized to values between 0 and 1 and all statistical analyses
were conducted in the R environment.

### 4 Results

In the following, the mean sample depth will be used to describe the position of each sample within the soil column.

### 4.1 Thaw depth

We found that thaw depth (July 2019) decreased with grazing intensity categories, giving a thaw depth of 38 cm below surface
(bs) at DB-IN, 59 cm bs in DB-EX and 85 cm bs in DB-NON with a semi-frozen zone between 80 and 85 cm bs. In UP-IN, we measured a thaw depth of 53 cm bs.



## 4.2 Vegetation

At DB-IN, located in an occasionally flooded area of a partially drained thermokarst basin, grassland dominated by Poaceae was the main vegetation type. Shorter grasses found in DB-IN exhibited signs of grazing by large herbivores.Additionally,
small patches of faeces and disturbed bare ground were present. These disturbed micro-sites supported the growth of forbs. DB-EX was located in a grassland with lower-growing Poaceae interspersed with forbs. This site was located close to the edge of the grazing area and was less frequently visited by animals. Extensive grazing activity at DB-EX was reflected in slightly less intense signs of grazing on plant leaves and by the presence of some faeces. The vegetation in DB-NON was characterized by grasses (e.g. *Beckmannia syzigachne*, *Hordeum jubatum*), forbs (e.g. *Saxifraga* sp.), *Equisetum* sp. and low-growing shrubs
(e.g. *Salix* sp.). At least sporadic herbivore presence at this site was indicated by faeces. Signs of ground disturbance were likely also human-induced by clearing of forests. DB-NON was located on the edge of the thermokarst basin and outside the fenced area with animals introduced only recently. It was previously covered in larch forest until 2015 before being cut clear to extend pastures.

In contrast to these sites, UP-IN was located on the uplands surrounding the thermokarst basin. Its vegetation was much more
structured in height, with low and tall shrubs (e.g. *Salix* sp.) up to 2.5 m height, and low vegetation (diverse forbs, grasses, *Equisetum* sp.) (10-20 cm height) in between. Signs of grazing evident especially on grasses, and faeces as well as small trampling disturbances were visible on the ground.

## 4.3 Carbon parameters

### 4.3.1 Total organic carbon (TOC)

At DB-IN, TOC overall decreases from top to bottom with the highest value of 25.66 wt% at 20.25 cm bs and lowest value of 1.18 wt% at 92.5 cm bs (Fig. 2). In contrast to this, TOC values for DB-EX peak around the frozen-unfrozen interface with values between 4.89 wt% (21.25 cm bs) and 30.10 wt% (64 cm bs) while the frozen part contains generally more TOC. The highest TOC values among all sites have been found in a peat layer at DB-NON in the frozen core part, with a peak value of 52.80 wt% (92.5 cm bs). The unfrozen core part contains much less organic carbon with values between 0.79 wt% (35.25 cm
bs) and 9.65 wt% (3.75 cm bs).

At the upland site UP-IN, TOC was higher in the unfrozen core part with values between 3.52 wt% (11.25 cm bs) and 10.73 wt% (49 cm bs). In the frozen core part below, TOC values were rather homogenous, varying between 1.01 wt% (103.5 cm bs) and 2.65 wt% (58.5 cm bs). In contrast to the previous sites, TOC at UP-NON is homogenous throughout the core at values between 1.24 wt% (33 cm bs) and 2.54 wt% (27 cm bs) with a slight increase at the bottom to 4.50 wt% (71 cm bs).





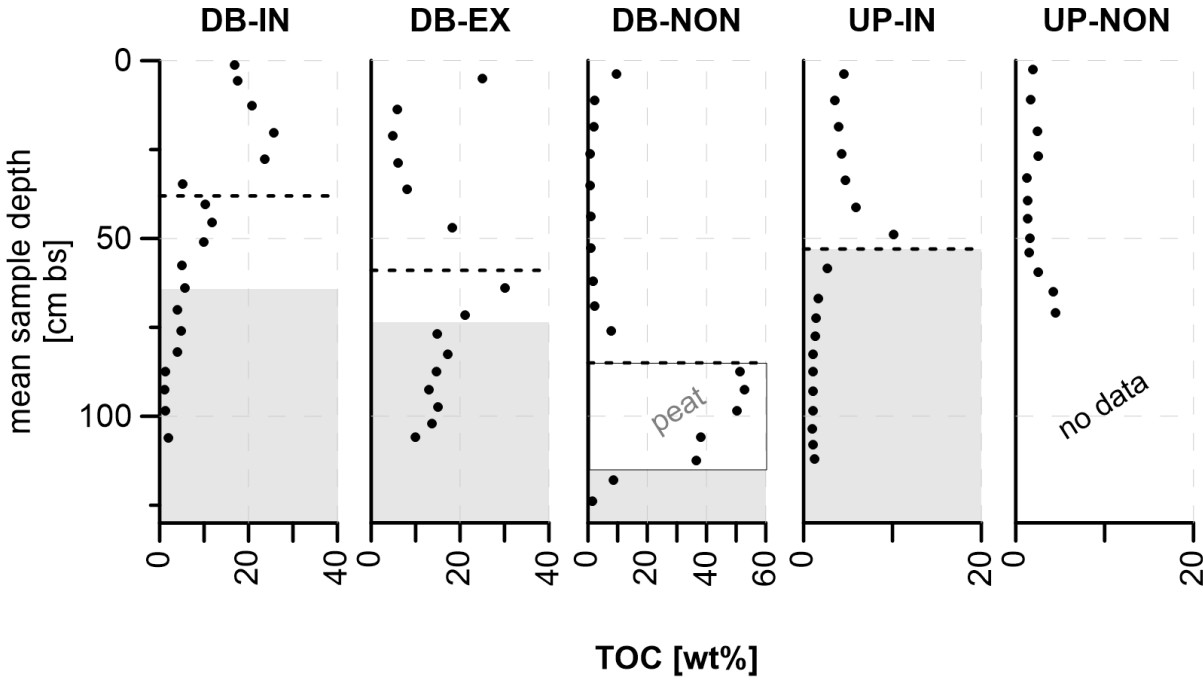

**Figure 2 – TOC values of all study sites; dashed lines mark the thaw depth found in July 2019; grey areas are assumed to be permafrost based on cryostratigraphic characteristics.**

### 4.3.2 TOC/TN and δ¹³C ratios

While C/N ratios for DB-IN (range 12.68 to 25.96) and DB-EX (range 14.83 to 18.48) are very similar (Fig. 3), values for DB-NON are higher, in the range of 16.94 to 24.90. In the upland sites, there is a strong contrast between UP-IN (11.40 to 29.19, mean of 17.75) and UP-NON (28.63 and 29.27) in C/N ratios.

In δ¹³C values, again the DB sites are very similar (DB-IN: -30.63 ‰ vs. VPDB (uppermost sample) to -26.43 ‰; DB-EX: -29.60 ‰ to -28.13 ‰; DB-NON: -30.17 ‰ to -27.89 ‰) (Fig. 3). Upland sites show higher δ¹³C values in the range of -28.06 ‰ to -23.49 ‰. For full δ¹³C values, please see figure S4.



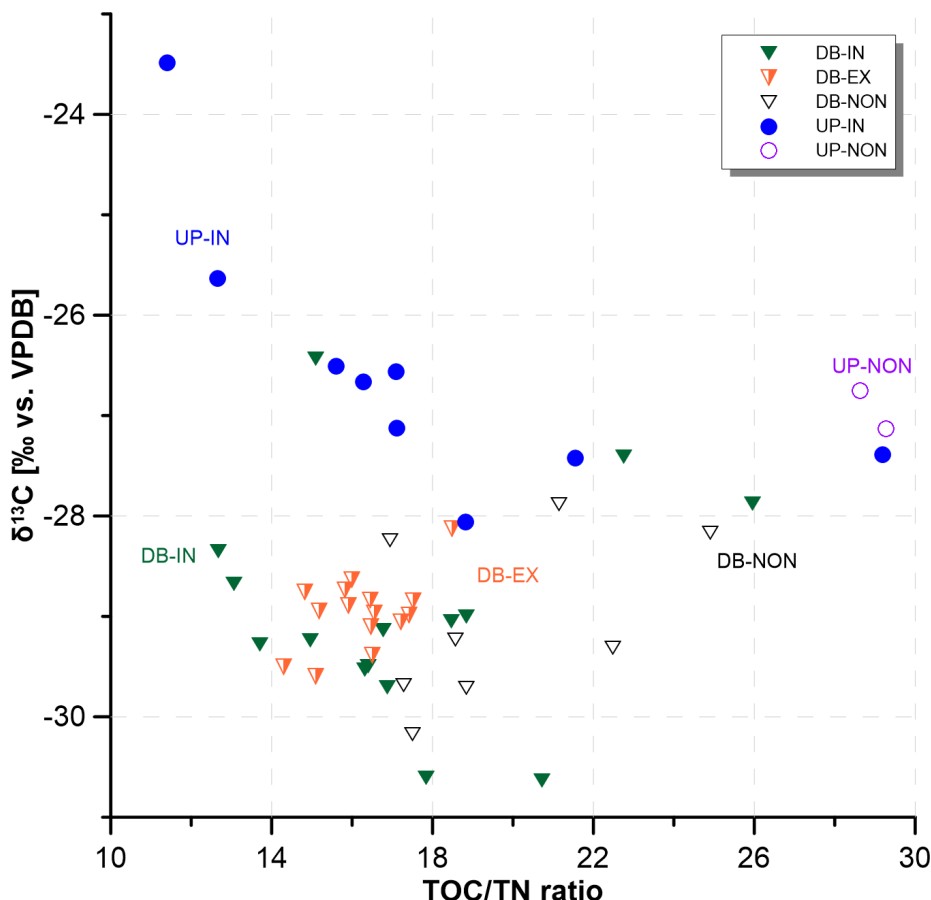


**Figure 3 - TOC/TN ratios plotted over δ¹³C ratios; main clusters of each sampling site are identified by circles.**

### 4.4 Radiocarbon ages

Radiocarbon ages for the DB sites ranged between 11,000 calibrated years before present (cal yr BP) and modern ages. At DB-IN, the age at the core's bottom (106 cm bs) is 10,568 cal yr BP (Fig. 4, table 1). Radiocarbon ages become younger higher

up in the core, reaching 111 cal yr BP at 27.75 cm bs. In DB-EX, the contrast between core bottom (3,616 cal yr BP at 106 cm bs) and top (2,038 cal yr BP at 21.25 cm bs) is less sharp. The oldest sample of DB-NON was dated to 5,154 cal yr BP (124 cm bs, core bottom) while a modern sample was found at 26.25 cm bs. We determined a basal peat age of 4,327 cal yr BP at 112.5 cm bs in DB-NON. The sediment sample directly above the peat layer found in DB-NON, in the unfrozen core part, was dated to 3,761 cal yr BP (76 cm bs). This gives an accumulation time of 566 years for approximately 30 cm of peat.

We found another sample containing modern material in UP-IN at 26.25 cm bs. The lower part of this core is considerably older than the other cores, reaching a maximum carbon age of 34,563 cal yr BP at 112 cm bs (core bottom). A sample taken from UP-IN at 82.5 cm bs was dated to 31,779 cal yr BP, while at 58.5 cm bs an age of 9,588 cal yr BP was found. These two



much older ages result from the dating of bulk material, most likely from Yedoma deposits, contrary to all other radiocarbon samples in this study, for which plant remains were dated.

The two radiocarbon samples obtained from UP-NON were dated to 7,278 cal yr BP (71 cm bs) and 183 cal yr BP (27 cm bs). For full details please see table 1.

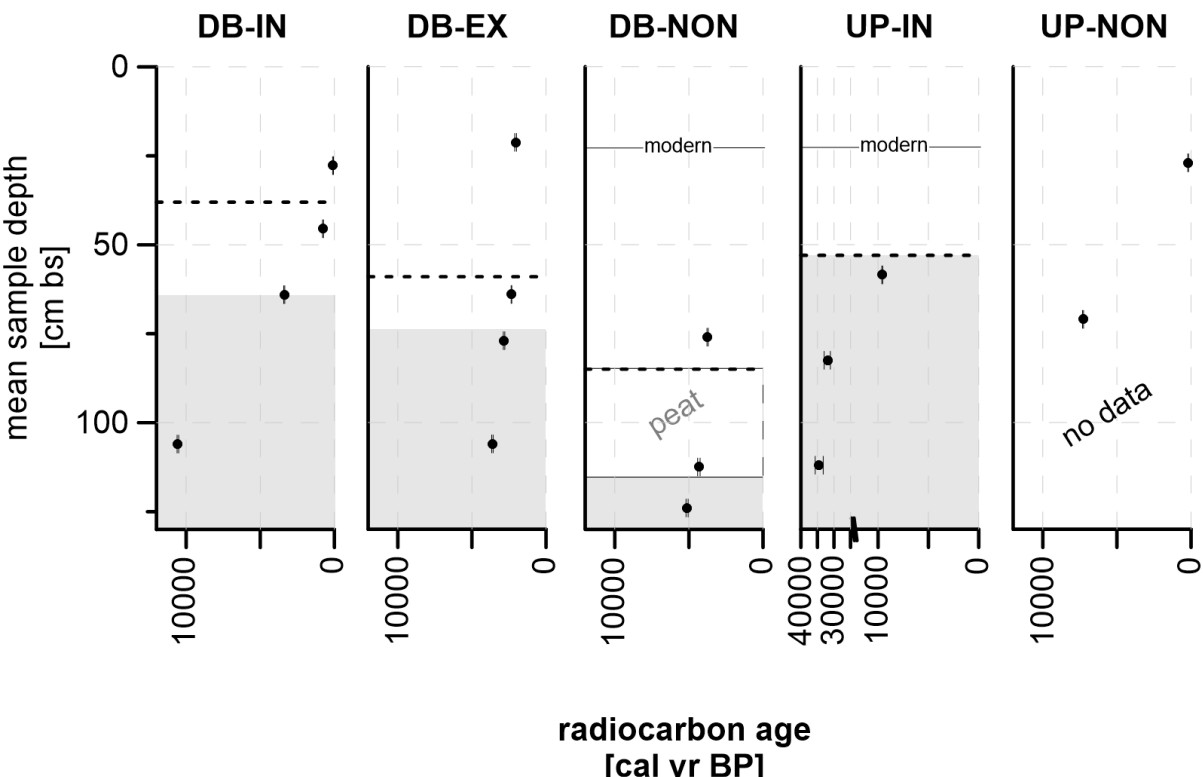

**Figure 4 - Radiocarbon ages determined for selected samples of each site, plotted over depth; please also see table 1; dashed lines mark the thaw depth found in July 2019; grey areas are assumed to be permafrost; note differently scaled**

**x-axes.**

### 4.5 Sediment parameters

We find the material to be homogenous throughout all study sites regarding grain size distribution (Fig. 5) and silty overall (Fig. S2 and S3). In DB-IN, clay content slightly decreases with depth from 16.69 vol% at 76 cm bs to 7.50 vol% at 106 cm bs, while at the same time sand content increases from 3.46 vol% to 10.68 vol%. Clay content varies throughout DB-EX with

a minimum of 14.16 vol% at 28.75 cm bs and a localised maximum of 22.17 vol% at 13.75 cm bs (Fig. 5). Sand content decreases with depth, starting at 12.73 vol% (5 cm bs) and reaching 2.43 vol% at the bottom (106 cm bs). In DB-NON, clay content shows a peak value of 21.96 vol% at 76 cm bs, followed by a peat layer down to 106 cm bs with no grain size data





available. Mean clay content for this core is 10.84 vol%. Sand content varies between 5.37 vol% (35.25 cm bs) and 11.31 vol% (43.75 cm bs) with two low values, 2.54 vol% at 52.75 cm bs and 2.22 vol% at 124 cm bs.

UP-IN shows similar characteristics as DB-IN with clay content decreasing with depth (highest value 19.76 vol% at 49 cm bs, lowest value 6.34 vol% at 112 cm bs), while simultaneously the sand content increases to a peak value of 11.52 vol% at 103.5 cm bs (Fig. 5). Grain size composition for UP-NON shows no peaks, with clay contents between 8.03 vol% (71 cm bs) and 12.60 vol% (33 cm bs) and sand contents between 8.20 vol% (44.5 cm bs) and 14.01 vol% (71 cm bs).

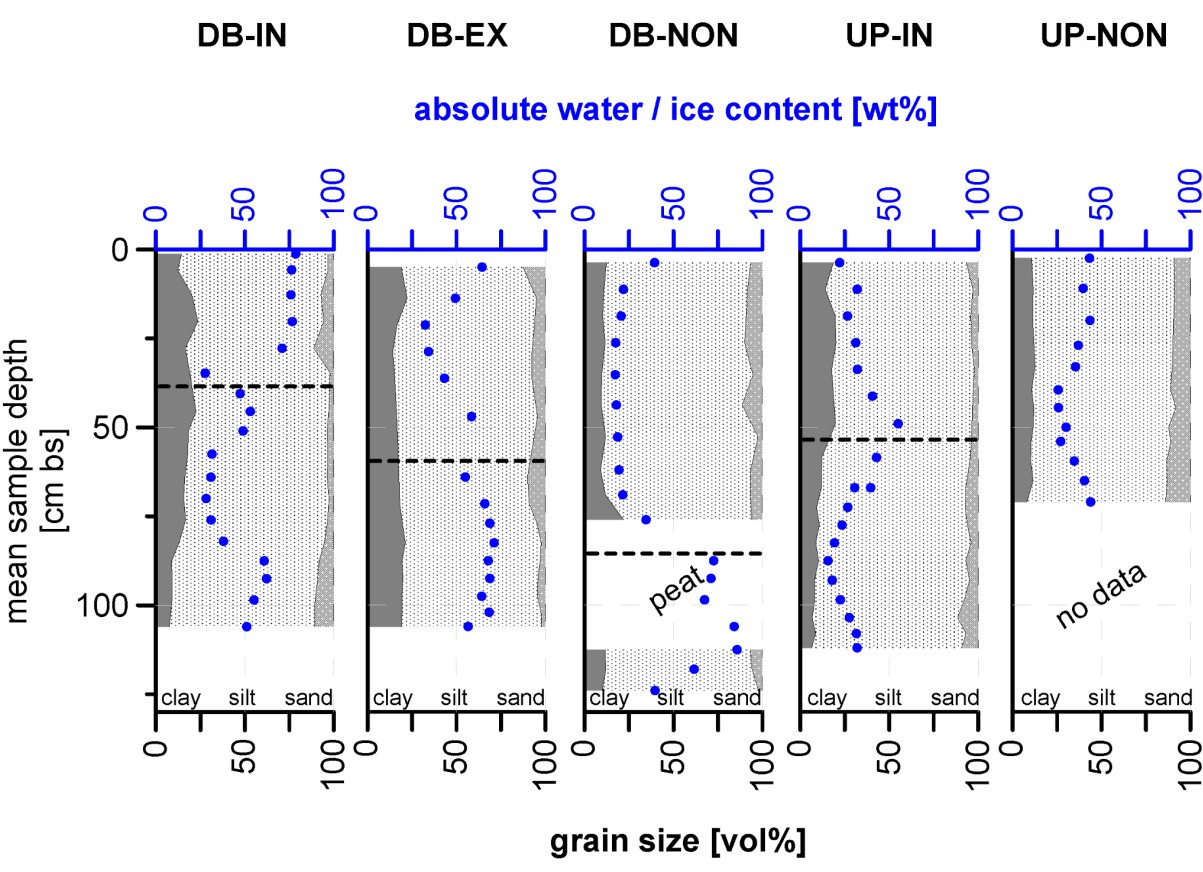

**Figure 5 - Grain size distribution for all sampling sites plotted over depth; absolute water and/or ice content added in blue; dashed lines mark the thaw depth found in July 2019.**

The absolute water and/or ice content, derived from weighing pre- and post-drying, shows generally more dry conditions in the frozen core parts in DB-IN, with values fluctuating between 27.67 wt% (34.75 cm bs) and 78.59 wt% (1.25 cm bs). Similar characteristics are present in DB-EX (34.26 wt% at 28.75 cm bs to 71.10 wt% at 82.5 cm bs). In contrast to that, DB-NON

shows large differences between the unfrozen upper part (17.19 wt% at 35.25 cm bs, 39.36 wt% at 3.75 cm bs) and the frozen lower part (39.59 wt% at 124 cm bs, 85.74 wt% at 112.5 cm bs).

UP-IN shows similar water content values at the top (32.00 wt% at 11.25 cm bs) and bottom (31.93 wt% at 112 cm bs) with higher values around the freezing interface (55.01 wt% at 49 cm bs) and lower values above and below that (15.59 wt% at 87.5 cm bs). Water content in UP-NON is more stable, ranging between 25.63 wt% (39.5 cm bs) and 43.88 wt% (71 cm bs) with a similar value at the top (43.24 wt% at 2.5 cm bs).

Stable isotope characteristics of the pore water are shown in supplement figure S5.

**4.6 Data analysis**

The PCA revealed a correlation between wetness of the site and TOC content (Fig. 6). While DB-IN, DB-EX, UP-IN and UP-NON form clusters, DB-NON values are spread across parameters. Especially for DB-IN and DB-EX we see a strong correlation between TOC and water/ice content. For full PCA scores, see table S1 as well as figure S1.

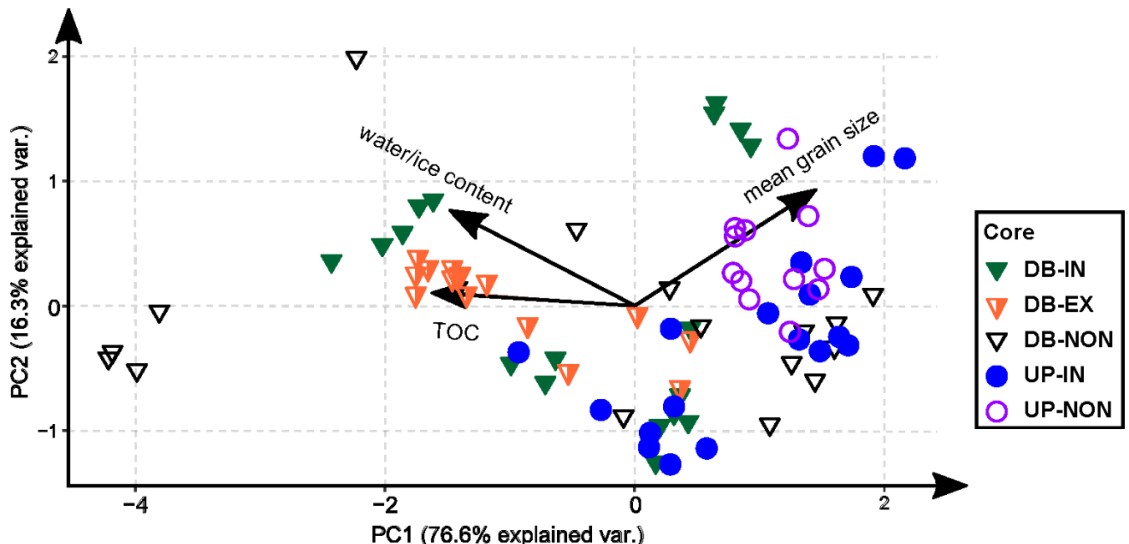

**Figure 6 - Results of the principal component analysis for variables TOC, water/ice content and mean grain size.**

A test for significance (Kruskal-Wallis H test) revealed no significant differences between the TOC values of intensively grazed sites in comparison to non-grazed sites (p-value of 0.6873). However, we found significant differences when comparing the uppermost 38 cm TOC values. Here we obtained a p-value < 0.001, which shows that differences in TOC between intensively and non-grazed sites are highly significant in the upper part of the soils, well within the active layer in all sites (Fig. 7).



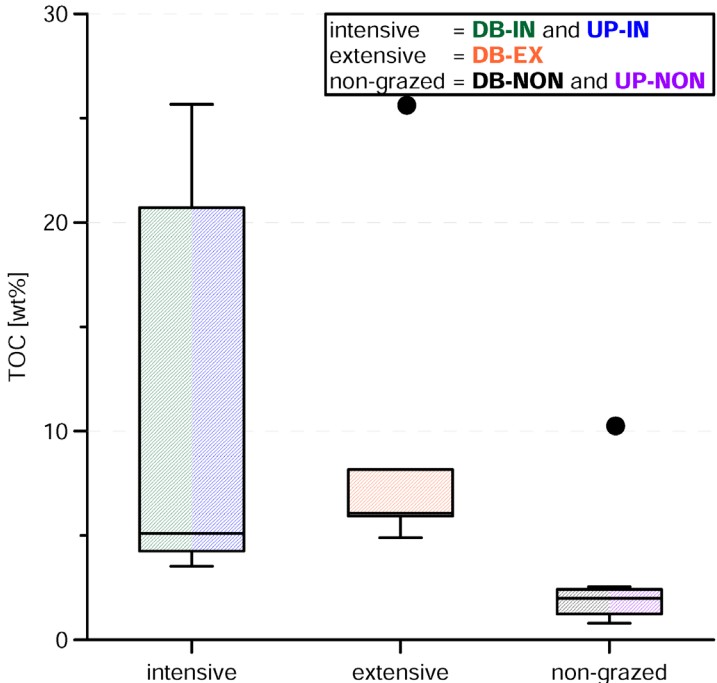

**Figure 7 - Boxplots for TOC data of all intensive, extensive and non-grazed sites compiled for the uppermost 38 cm of each core; sites DB-IN and UP-IN are combined in the "intensive" boxplot, DB-EX is shown in the "extensive" boxplot and DB-NON and UP-NON are combined in the "non-grazed" boxplot; the median is shown by a vertical line within each box; box margins show the upper and lower quartile; whiskers mark minimum and maximum values; outliers (indicated by dots) are more than 1.5 box lengths away from box margins.**

## 5 Discussion

### 5.1 Effects of grazing on vegetation structure and permafrost thaw

We found that intensively grazed sites (DB-IN, UP-IN) are covered by generally shorter and more sparse vegetation (Fig. S2 DB-IN: grassland; Fig. S3 UP-IN: shrubby grassland tundra), with taller grazing-resistant individuals in between, compared to extensively or non-grazed sites (Fig. S2 DB-EX: grasses and herbs, and DB-NON: grasses, herbs and low shrubs). This could be either a result of the reduction in shrub expansion through grazing (Suominen and Olofsson, 2000), or of the differences in the flooding regime between the sites, which promotes different vegetation types: at least the grazed sites located in the thermokarst basin (DB-IN and DB-EX) are still flooded seasonally, and the upland sites are generally better drained than the sites within the basin. We aimed at selecting representative sampling locations, based on overall vegetation and observed animal routes. However, we have little information on representativeness in terms of soil type and wetness.




Overall, our results suggest that the changes in vegetation height and structure are a result of animal grazing, in agreement
with studies in similar Arctic settings (Sundqvist et al., 2019; Verma et al., 2020; Skarin et al., 2020), although the effect of
seasonal flooding on vegetation composition and structure in some of the sites (DB-IN and DB-EX) is likely very important
too.

For our study area, shrubs established on the upland before the introduction of large numbers of animals, and are now retreating
due to grazing pressure. On the contrary, at the edges of the drained thermokarst basin, shrubification takes place on spots only
extensively grazed, following the retreating water line caused by lake drainage and hence vegetation establishment. This leaves
us with high soil wetness as another plausible explanation for the absence of shrub vegetation.

The limited grazing space within the fenced Pleistocene Park area leads to a higher revisiting rate of animals for sites within
the fence. This artificially increased grazing pressure likely led to the different vegetation structures observed inside and
outside of the fence.

Active layer thickness is associated with warming of the ground, influenced by summer temperatures, insulation from snow,
and vegetation density and composition (Walker et al., 2003; Skarin et al., 2020). Therefore, following Zimov et al. (2012),
influencing these insulating factors should affect the active layer depth by allowing for stronger cooling of the ground in winter
(Fig. 8). However, to determine any trends, repeated annual measurements of the thaw depth are needed.

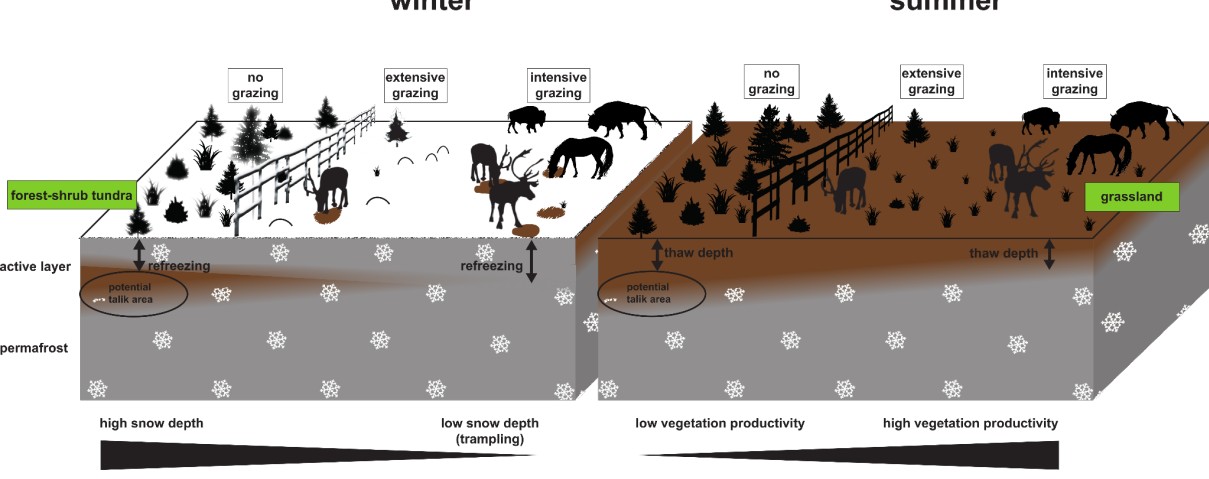

**Figure 8 – graphical representation of S. Zimov's hypothesis; winter: greater numbers of large animals trample down
or partly remove the snow cover, facilitating the full refreezing of the active layer; summer: greater numbers of animals
alternate the vegetation type from shrub areas towards grassland, increasing vegetation productivity; as the active
layer refreezes in winter in the grassland areas, thaw is reduced compared to non-grazed tundra and forest areas.**

We see from the DB sites data that the more intensively grazed area featured smaller thaw depths in July 2019, compared to
the less or non-grazed sites (Fig. 2). Looking into the assumed permafrost table depth based on cryostratigraphic characteristics
in the cores, we observe the same effect with an active layer depth of 61 cm bs under intensive grazing at DB-IN, 74 cm bs



under extensive grazing at DB-EX and 115 cm bs for the non-grazed DB-NON site. Despite results being consistent with the expected effects of grazing on ground temperature, the limited core replication of the present study combined with the expected spatial heterogeneity of active layer depths means that this interpretation needs caution. Further, more replicated studies, potentially combined with spatially comprehensive modelling exercises, are advised.

At DB-IN, the grasses are thought to lay flat beneath the snow cover. The snow itself is compacted by animal trampling, thereby further reducing the insulation properties of the snow cover mostly due to snow densification. The forb undergrowth of the grassland at DB-EX provides only slightly more resistance against the weight of the snow cover, but combined with less intensive trampling, this site would have a less compact snow cover with better insulation properties for DB-EX compared to DB-IN. The *Salix* shrubs at DB-NON promote localized snow trapping and a loose snowpack, both of which have been shown to provide effective insulation against winter cold (Myers-Smith et al., 2011). Trampling is also minimal in this ungrazed area. In addition to those surface conditions, the composition of the sediment itself could contribute to deeper active layers at site DB-NON. From grain size distribution data (Fig. 5), we see that material among all studied sites is quite homogeneous. Exceptions are the DB-NON and UP-NON sites, which contain a smaller share of clay-sized particles, leading to a coarser overall material. High ice contents in the frozen part of the DB-NON core support the idea of an insulating effect of the peat layer that prevents the ground from thawing any deeper.

For the upland sites, we were not able to compare thaw depths: the cryostratigraphically-determined permafrost table depth of approximately 50 cm bs at UP-IN is, however, smaller than active layer depths provided by Abramov et al. (2019) for the close-by Mt. Rodinka site, located on a Yedoma upland with a mean active layer depth of 80 cm. Also evident in this study are smaller thaw depths around 40 cm for alas sites featuring shorter vegetation in the Kolyma region (Abramov et al., 2019). It is most likely that the shorter graminoid vegetation types are linked to grazing intensity (Forbes, 2006). Combined with trampling of snow cover, this leads to colder ground conditions, hence stabilizing permafrost. Other effects coincidentally aligning in the same pattern, such as e.g. soil moisture content, as visualized by the PCA (Fig. 6), could not be excluded and have to be tested in other locations. However, the contrast in the decomposition state (TOC/TN ratios and $\delta^{13}$C ratios; Fig. 3) of the studied deposits between intensive and non-grazed locations suggests that animal presence is an important driver in ground carbon storage. This favours the grazing hypothesis as a more likely explanation for the significantly higher TOC content in the grazed and trampled locations.

## 5.2 Carbon accumulation under grazing impact

We found that the TOC content is on average six times higher in the top 38 cm of DB-IN compared to DB-NON (Fig. 2 and 7). DB-EX as the intermediate state reaches TOC values three times higher than DB-NON. For the upland sites, TOC is twice as high under intensive grazing in the top core parts than in non-grazed cores. The effect is not visible in lower core parts, perhaps because of the relatively short time span of 23 years since intensified animal introduction into the area. This points out that animal grazing can indeed increase carbon storage in a relatively short time span, if there are ways to incorporate OM into the ground, such as unfrozen conditions. However, these differences are not significant when comparing the whole core,



which we explain as a result of the TOC-rich peat layer found in DB-NON that shifts the TOC value median for non-grazed sites upwards, combined with low TOC values in the frozen part of the intensively grazed sites that is in strong contrast to the active layer here. Since this peat layer is found at 85 cm bs, we do not relate it to herbivory influence in a 23 year timespan, which is why we tested for significance in the active layer again.

In general, there are several ways for carbon accumulation in permafrost areas, such as deposition of organic-rich sediments
(via aeolian or fluvial transport, Chlachula (2003); Huh et al. (1998)), increased in situ biomass accumulation (via increased plant growth, Schuur et al. (2008)), cryoturbation (Kaiser et al., 2007) or also animal influence via feces. The last is also linked to increased plant growth by providing easily available nutrients (Grellmann, 2002). Also the disturbance of the surface layer via trampling mixes fresh OM into the ground, providing additional OM input that adds to the previously permafrost-preserved OM. The significant difference in upper soil TOC agrees with the expected effects of grazing, as it is perfectly in line when
comparing sites of intensive large herbivore presence (DB-IN and UP-IN) with non-grazed sites (DB-NON and UP-NON).

All organic carbon found is less decomposed in the upland areas (higher $\delta^{13}$C values, Fig. 3), which is a result of colder, drier and more compact ground conditions of the Yedoma deposits, compared to the seasonally flooded and less compact ground of the drained thermokarst basin. This supports the hypothesis of generally colder ground conditions due to intensive grazing leading to reduced OM decomposition (Aerts, 2006). Higher C/N ratios in non-grazed sites indicate a different source material
from shrubby tundra, in comparison to grassland vegetation in extensively and intensively grazed sites.

We found older carbon (2,338 cal yr BP) in the active layer part in DB-EX (Fig. 4). This hints at a recent deepening of the active layer, with old OM now becoming subject to decomposition. In DB-IN, a radiocarbon age of 789 cal yr BP was determined at 45.5 cm bs. However, when comparing both DB-IN and DB-EX to the DB-NON site, where modern material was found at 26.25 cm bs, this supports our hypothesis of higher and probably increased carbon storage under animal grazing
pressure as this material was likely mixed into the ground by animal trampling.

Stable permafrost conditions are also visible from the radiocarbon data of UP-IN, where the radiocarbon age at 58.5 cm bs, shortly below the active layer, is 9,588 cal yr BP, while modern material was found at 26.25 cm bs. This shows a mixing of the active layer, probably caused by animal influence or also cryoturbation, with OM in the permafrost being stored under stable conditions at the same time. However, it is possible that the active layer was shallower in the past and is now deepening
towards older material.

In the low and carbon-poor parts of the UP-IN core the sediment was dated to >30,000 cal yr BP (Fig. 4) and in a rather undecomposed state (Fig. 5), indicating a long-term frozen state of this site.

Two points can be summarized from these carbon data:

(1) In the upper part of the soil cores, TOC contents are higher in more intensively grazed sites, with a decrease along the
reduction in grazing intensity for both thermokarst-affected and original Yedoma sites. As the non-grazed UP-NON Yedoma site matches findings from other Yedoma studies (Strauss et al., 2017; Jongejans et al., 2018; Windirsch et al., 2020), we consider this as the original (i.e., not heavily grazed) state, showing that intensive grazing for 23 years already increased TOC



amounts stored in the active layer by factor two (Fig. 2). At thermokarst sites, hydrology has an impact on TOC storage as well, as the strong correlation between TOC content and water content is visible from the PCA (Fig. 6).

(2) The much shallower thaw depth in intensively grazed areas is the second important finding here. It shows that promoted ground cooling effects resulting from vegetation shifts towards graminoid communities as well as snow compaction by animal trampling can alternate the ground thermal regime and therefore active layer depth during a rather short period of time. While TOC is significantly higher in the active layer in intensively grazed sites, the shallower thaw depths associated with these intensively grazed sites stabilize underlying permafrost and hence prevent permafrost-stored OM from further degradation.

## 6. Conclusion

By analyzing a set of parameters for permafrost and active layer soils obtained from sites with different grazing intensity in both a drained thermokarst basin and a Yedoma upland in northeastern Siberia, we found evidence in favor of the hypothesis that intensive grazing by large herbivorous animals led to increased carbon storage in our studied permafrost sites over a 23 year time period. At the same time, annual thaw depth is lower at intensively grazed sites. These changes of ground characteristics are likely a combined result of vegetation changes and snow insulation reduction, probably amplified by additional carbon input due to intense herbivore impacts. Vegetation appears to change from shrubby tundra to grasslands under herbivory impact in our study area.

We conclude that intensified animal husbandry could therefore effectively be used to stabilize and even increase carbon storage at non-forest permafrost sites, a strategy that may fit into a broader set of instruments mitigating climate change consequences. To further investigate these effects it is necessary to sample grazing intensity transects in higher spatial resolution and repeat this approach in other permafrost areas as well. Also, longer time series and monitoring approaches are needed. For studies on potential use of animal grazing as a measure against permafrost thaw, a pan-Arctic network of similar experimental sites would be necessary.

### Data availability

All measurement data are available via PANGAEA under DOI 10.1594/PANGAEA.933446 (Windirsch et al., 2021).

### Author contribution

TW, GG, MU and JS designed the study. TW conducted field and laboratory work, prepared the graphics and led the writing of this manuscript. TW, GG, MU and JS analyzed and interpreted the laboratory results. GG designed the maps used in this study. BF, MM-F and JO provided expertise on herbivory and herbivore-environment interactions. JW contributed expertise in vegetation classification and statistics. NZ and MG provided expertise on the area and local environment processes and characteristics as well as on the Pleistocene Park experiment. All authors contributed to compiling and editing the manuscript.



**Competing interests**

The authors declare no conflict of interests.

**Acknowledgements**

We thank the Field Experiments & Instrumentation team at the Max Planck Institute for Biogeochemistry in Jena as well as Juri Palmtag for helping with the drilling campaign. We further acknowledge Dyke Scheidemann, Jonas Sernau and Angelique Opitz (Carbon and Nitrogen Lab [CarLa]) as well as Mikaela Weiner and Hanno Meyer (Stable Isotope Lab) from AWI for assistance in the laboratory. We thank Christian Knoblauch (Universität Hamburg) for helping with TOC measurements. This study was supported by the Northeast Science Station team in Chersky, Sakha. We further thank J. Otto Habeck (Universität

Hamburg) for his help in designing this study.

The field campaign was carried out in the framework of the CACOON project (#03F0806A (German Federal Ministry of Education and Research)) and as part of the PeCHEc (Permafrost Carbon Stabilization by Recreating a Herbivore-Driven Ecosystem) project funded by the Potsdam Graduate School. The authors received additional support from the Geo.X research network (SO_087_GeoX). Additional funding was provided by the CHARTER project (grant agreement ID 869471).

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



**Table 1 - Radiocarbon measurement data and calibrated ages**

| Site | Mean sample depth [cm bs] | material | $^{14}$C age [yr BP] | +/- [yr] | F$^{14}$C | +/- [abs] | Calibrated ages (2 σ)* [cal yr BP] | Mean age [cal yr BP] | AWI no. |
|---|---|---|---|---|---|---|---|---|---|
| DB-IN | 27.75 | plant/wood | 141 | 16 | 0.9826 | 0.0020 | 58 - 118 | 111 | 6742.1.1 |
| | 45.5 | plant/wood | 899 | 18 | 0.8941 | 0.0020 | 733 - 801 | 789 | 6743.1.1 |
| | 64 | plant/wood | 3157 | 18 | 0.6750 | 0.0015 | 3350 - 3412 | 3384 | 6744.1.1 |
| | 106 | plant/wood | 9356 | 24 | 0.3120 | 0.0009 | 10499 - 10609 | 10568 | 6745.1.1 |
| DB-EX | 21.25 | plant/wood | 2081 | 17 | 0.7717 | 0.0017 | 1993 - 2110 | 2038 | 6746.1.1 |
| | 64 | plant/wood | 2300 | 17 | 0.7510 | 0.0016 | 2311 - 2351 | 2338 | 6747.1.1 |
| | 77 | plant/wood | 2765 | 18 | 0.7088 | 0.0016 | 2782 - 2886 | 2854 | 6748.1.1 |
| | 106 | plant/wood | 3382 | 18 | 0.6564 | 0.0014 | 3569 - 3664 | 3616 | 6749.1.1 |
| DB-NON | 26.25 | plant/wood | modern | | 1.0263 | 0.0020 | | | 6750.1.1 |
| | 76 | plant/wood | 3487 | 18 | 0.6479 | 0.0014 | 3695 - 3781 | 3761 | 6751.1.1 |
| | 112.5 | plant/wood | 3879 | 18 | 0.6170 | 0.0014 | 4242 - 4407 | 4327 | 6752.1.1 |
| | 124 | plant/wood | 4533 | 20 | 0.5688 | 0.0014 | 5052 - 5188 | 5154 | 6753.1.1 |
| UP-IN | 26.25 | plant/wood | modern | | 1.0262 | 0.0020 | | | 6754.1.1 |
| | 58.5 | plant/wood | 8643 | 23 | 0.3410 | 0.0010 | 9539 - 9633 | 9588 | 6755.1.1 |
| | 82.5 | bulk | 27748 | 417 | 0.0316 | 0.0016 | 31105 - 32971 | 31779 | 6756.1.1 |
| | 112 | bulk | 30099 | 563 | 0.0236 | 0.0017 | 33267 - 35673 | 34564 | 6757.1.1 |
| UP-NON | 27 | plant/wood | 191 | 16 | 0.9766 | 0.0019 | 162 - 218 | 183 | 6758.1.1 |
| | 71 | plant/wood | 6355 | 21 | 0.4533 | 0.0012 | 7251 - 7323 | 7278 | 6759.1.1 |

* calibrated using Calib 8.2 (Stuiver et al., 2021) equipped with IntCal20 (Reimer et al., 2020)