# Peer review of "Large Herbivores Affecting Permafrost - Impacts of Grazing on Permafrost Soil Carbon Storage in Northeastern Siberia"

_Biogeosciences, 2021_

## Author Comment (AC1)

**Authors' reply to RC #2 on bg-2021-227**

Author's reply to RC #2 on "Large Herbivores Affecting Permafrost – Impacts of Grazing on Permafrost Soil Carbon Storage in Northeastern Siberia" by Torben Windirsch et al., Biogeosciences Discuss., https://doi.org/10.5194/bg-2021-227, 2021

Authors' replies are formatted in blue.

This paper entitled "Large herbivores affecting permafrost – impacts of grazing on permafrost soil carbon storage in northeastern Siberia" by Windirsch and others reports carbon storage in the active layer of an intensively grazed thermokarst basin in Siberia. The study tests the important hypothesis of whether reintroduction of large mammals could slow permafrost degradation and greenhouse gas release with climate change. They found decreased active-layer depth and increased carbon storage in the intensively grazed location. The paper is very well written, and the hypotheses are well conceived and explained. I have some questions and concerns about the experimental design, which may simply be resolved with clearer description, or they might represent more serious issues.

Thank you for acknowledging the importance of our study. Following your advice, we will clarify the points raised below in the revised manuscript.

- Comparing the effect size with expected carbon release would be very helpful in interpreting the implications of this study. For example, how much of the permafrost carbon feedback could be reduced were large areas of the permafrost zone managed in this way? I know there are large uncertainties involved in this kind of analysis, but given the importance of this issue (and the hype and criticism this particular location receives) a "back of the envelope" calculation of potential importance seems justified.

Thank you. As you mentioned, uncertainties are too high to make thus semi-quantitative back on the envelope calculations. Also, we do not have emission data from our sites, which would be needed to reduce these uncertainties. Anyway, we will improve the section by adding a qualitative discussion of scalability and other challenges.

- As currently written, the grazing intensity seems to have only been assessed qualitatively (indicated by park staff). This could introduce several nonrandom sources of error. For example, the animals are very likely to prefer certain ecosystem types, which could cause a strong difference in soil temperature and carbon content independent of the effect of grazing. Additionally, the assessment of the park staff could have been influenced by their knowledge of the intent of the experiment. In a non-blind assessment, this kind of implicit bias is common in qualitative assessments.

Thank you for this hint. We spent several days in the park selecting our study sites while observing the animals, their paths and feeding places. We will clarify this in the manuscript. Also, it is true that animals prefer specific ecosystems. However, in this rather small and fenced setting their choice is very limited. While grazing, they actively change the ecosystem, especially vegetation, at the sites with highest grazing intensity, and apparently prefer that ecosystem type they help creating. So in conclusion, this is adding to the idea of long-term landscape changes by herbivory. You are right this is a qualitative start, a stratified random sampling approach would be perfect for studies after this pilot study, but due to logistical and budgetary constraints, this was not possible for this study

- Given the non-split-plot design, is it possible pre-existing differences in carbon stocks and soil thermal dynamics account for some of the observed difference? There can be high spatial heterogeneity in locations such as this—particularly given the fluvial and cryostratigraphical differences that are suggested by the satellite images. The differences in radiocarbon age estimates among the sites suggest to me that they were

not similar to begin with—casting some doubt on the conclusion that the grazing treatment accounts for the observed differences. This could perhaps be addressed by comparing results here to a larger suite of measurements from the cited studies in this paper. For future work, using exclosures within the grazed area would allow direct testing of cause and effect with both soil and vegetation.

We agree that hydrology and cryostratigraphy might also be contributing to the found differences, and will emphasize this option in the revised manuscript. However, since the differences in active layer depth and carbon content between high grazing intensity and no grazing are similar in two different landscape forms – where hydrology and cryostratigraphy are different – we conclude that herbivory is at least contributing to these differences as well. We do not have any information about the initial state pre-experiment, but that is why we chose to sample two different landscape types – upland and basin – to see if the direction of development under grazing influence is similar. Since this is only a pilot study to test for methodology, we did not set up any exclosure sites but decided to treat the area outside the fence as an exclosure. Indeed, exclosure sites would be very helpful in the future but also require a long-term establishment phase.

- How much of the observed change could be due to bulk density effects? Large herbivore presence can cause compaction, which would result in potentially a decrease in the measured active-layer thickness even if the permafrost table absolute position moves downward.

Thank you for bringing this up. The substrate at these sites is silty-clayish sediment, topped by an organic layer with dense roots. We assume that compressibility is very low in that case. In peatlands, this effect would be very important, but the only peat we encountered was within the permafrost at a depth of 80 cm bs at the non-grazed basin site. We will include the effects of trampling compaction about different soils in our improved discussion

- The question of whether large herbivores could decrease the permafrost carbon feedback depends on the effect on net ecosystem carbon balance—including more than just soil carbon. When aboveground carbon stocks are taken into account, how does this influence the conclusions?

The aim of our paper was not to decipher the net ecosystem carbon balance, but to have a first look on how to keep as much carbon in the ground by slowing down the warming of permafrost.
For a net balance you are absolutely right, but this is beyond our aim. Thus, we did not look into aboveground carbon stocks. To comment on the effects on the permafrost carbon feedback, of course an assessment of aboveground carbon stocks is needed. However, in this study we focussed on the effects of herbivory within the ground and hesitate to comment on broader implications for the climate.

- The extensive use of acronyms to refer to treatments and sites added unnecessary complexity. I would recommend to use intuitive words rather than acronyms whenever possible.

Thank you, we will implement a new naming scheme.

Line edits:

39: Could you mention the percentage difference here to give readers an idea of the effect size?
Thank you for this idea, we will add this.

54: some additional references on this, including some counterintuitive vegetation-soil interactions (Kropp et al., 2020; Loranty et al., 2018; Mekonnen et al., 2021) Pertinent

findings from another herbivore manipulation in tundra: (Min et al., 2021; Strebel et al., 2010)
Thank you very much!

174: The C13 signature is influenced by many factors besides degradation state (Abbott et al., 2016; Malone et al., 2018; Mauritz et al., 2019).
Yes, we will include its potential meaning for source and degradation in the revised version.

212: What influence might this recent deforestation have had on soil thermal dynamics?
Thank you. Unless more shadow in the three to four summer months we expect the ground has likely cooled after deforestation. Reason for that is that the trees did not function as a natural snow fence anymore, which resulted in a thicker snow cover isolating against eight months of cold winter temperatures. However, we still encounter the greatest active layer thickness at this site, so it either was even deeper before deforestation, or the removal of the trees has – against our hypothesis – increased warming by eliminating the shading effect of the trees in summer. We will add this to the manuscript.

287: A more informative subsection name would be helpful.
Thank you, we will change this to "statistics".

300: What do the colors in the boxplots represent?
The colors are used to encode which sites are included in which box. Therefore, box colors match the site name colors in the legend.

330: This figure might be more appropriate in the introduction, since it depicts the hypothesis rather than the findings. If included here, I would recommend annotating to show the predictions that were confirmed, disproven, or inconclusive. One small note: it looks like both Wisent and American Bison are shown in this picture. Are both species present at this site?
Thank you, we will move this figure to the introduction. And yes, there are both species present at the park.

340-375: This section seems like an extension of the results rather than a contextualization or discussion. I might recommend shortening, adding references to other work, or moving the supplementary information.
Thank you for this comment, we will shorten this section and add references to make it more suitable for a discussion section.

380: The pattern fits, but the study design does not seem able to establish if this was due to grazing or not.
Yes, thank you, we will add the other possible causes here. Moreover we will adjust the title to make clear that this is a pilot study.

390: All this discussion of the radiocarbon age being attributable to active-layer deepening and potentially to the grazing treatment assumes that pre-treatment SOM content and age were similar across sites. Is there evidence from other nearby sites that profiles are similar enough to assume they were the same pre-treatment?
Unfortunately not. We are not aware of any similar findings close by. We will remove this paragraph.

References

Abbott, B. W., Baranov, V., Mendoza-Lera, C., Nikolakopoulou, M., Harjung, A., Kolbe, T., et al. (2016). Using multi-tracer inference to move beyond single-catchment ecohydrology. *Earth-Science Reviews*, *160*(Supplement C), 19–42. https://doi.org/10.1016/j.earscirev.2016.06.014

Kropp, H., Loranty, M. M., Natali, S. M., Kholodov, A. L., Rocha, A. V., Myers-Smith, I., et al. (2020). Shallow soils are warmer under trees and tall shrubs across Arctic and Boreal ecosystems. *Environmental Research Letters*, *16*(1), 015001. https://doi.org/10.1088/1748-9326/abc994

Loranty, M. M., Abbott, B. W., Blok, D., Douglas, T. A., Epstein, H. E., Forbes, B. C., et al. (2018). Reviews and syntheses: Changing ecosystem influences on soil thermal regimes in northern high-latitude permafrost regions. *Biogeosciences*, *15*(17), 5287–5313. https://doi.org/10.5194/bg-15-5287-2018

Malone, E. T., Abbott, B. W., Klaar, M. J., Kidd, C., Sebilo, M., Milner, A. M., & Pinay, G. (2018). Decline in Ecosystem δ13C and Mid-Successional Nitrogen Loss in a Two-Century Postglacial Chronosequence. *Ecosystems*, *21*(8), 1659–1675. https://doi.org/10.1007/s10021-018-0245-1

Mauritz, M., Celis, G., Ebert, C., Hutchings, J., Ledman, J., Natali, S. M., et al. (2019). Using Stable Carbon Isotopes of Seasonal Ecosystem Respiration to Determine Permafrost Carbon Loss. *Journal of Geophysical Research: Biogeosciences*, *124*(1), 46–60. https://doi.org/10.1029/2018JG004619

Mekonnen, Z. A., Riley, W. J., Berner, L. T., Bouskill, N. J., Torn, M. S., Iwahana, G., et al. (2021). Arctic tundra shrubification: a review of mechanisms and impacts on ecosystem carbon balance. *Environ. Res. Lett.*, 29.

Min, E., Wilcots, M. E., Naeem, S., Gough, L., McLaren, J. R., Rowe, R. J., et al. (2021). Herbivore absence can shift dry heath tundra from carbon source to sink during peak growing season. *Environmental Research Letters*, *16*(2), 024027. https://doi.org/10.1088/1748-9326/abd3d0

Strebel, D., Elberling, B., Morgner, E., Knicker, H. E., & Cooper, E. J. (2010). Cold-season soil respiration in response to grazing and warming in High-Arctic Svalbard. *Polar Research*, *29*(1), 46–57. https://doi.org/10.1111/j.1751-8369.2010.00154.x

---

## Author Comment (AC2)

**Authors' reply to RC #1 on bg-2021-227**

Author's reply to RC #1 on "Large Herbivores Affecting Permafrost – Impacts of Grazing on Permafrost Soil Carbon Storage in Northeastern Siberia" by Torben Windirsch et al., Biogeosciences Discuss., https://doi.org/10.5194/bg-2021-227, 2021

Authors' replies are formatted in blue.

This manuscript presents observations of soil cores from different landscape units and disturbance histories, and aims to answer an interesting and relevant question, "does grazing by large mammals impact permafrost carbon storage?" Unfortunately, the experimental design is fundamentally flawed, making any conclusions about the impact of herbivory on soil carbon storage impossible.

Thank you for acknowledging the importance of the research topic. We should initially have clarified that this was a pilot study. We will do so in the revised manuscript, and adjust the study's aim accordingly.

The main issue is lack of replication – the study relies on a single soil core for each combination of environment (drained lake basin or upland) and grazing (intensive or no grazing), which is insufficient given the variability of soil composition and the presence of confounding variables.

This lack of replications is a result of the pilot study character. Also because of logistical constrains we designed this study as small as possible. A random sampling design would be best, yes, and would be a next step. We aim with this study to identify differences between sites that could result from herbivory. This will be clarified.

We know that soil core properties are highly variable in permafrost environments due to cryoturbation, so any variation from one site to another could be due to natural spatial variability or the variable of interest, herbivory. Without replication within sites to account for spatial variability of permafrost soils there is no way to discern between those two possibilities.

We will adjust the paper and especially the discussion accordingly, stronger emphasizing that natural disturbances and variability are equally likely to cause the discovered differences. We will also include a discussion on the scalability of herbivory impacts

Additionally, soil moisture is a confounding variable that cannot be accounted for without additional samples in a wider range of environmental conditions. The authors showed that soil organic carbon varied with water/ice content and mentioned that the grazed sites in the drained lake basin flooded seasonally, while none of the other sites flood regularly. This means that patterns in soil organic carbon may be due primarily to variation in soil moisture rather than herbivory, because soil moisture and herbivory covary.

Thank you for clarification. However, the pattern of much higher carbon content in the active layer is consistent across two different landscape types, which hints on a process present in both landscape types. So there is a possibility that herbivory could be the driver here. Of course, we will clarify also the other influencing processes, and definitively soil moisture and hydrology are of utmost importance here.

Another potential confounding variable is the site history. The authors mentioned that the non-grazed drained lake site was cleared of forest a few years prior to the study while none of the other sites underwent the same treatment.

This clear-cut was done 4 years before the study, while the other sites did not feature any forest vegetation. We selected this site since changes from wooden plants - although not trees but shrubs – towards grasses are associated with herbivore activity, making this site something like "ground zero" for vegetation succession with no animal activity yet.

While the underlying soil core data are sound and could be used to describe some of the variability of the site, the flawed study design makes it impossible to disentangle the effects of spatial heterogeneity, soil moisture regime, site history, and herbivory. Therefore, I suggest that this manuscript be rejected and the authors reconsider the scope of question that can be answered with these data for a new submission.

Thank you for this detailed comment on our study design. We agree that replication as well as random sampling would indeed provide more certainty.

We designed our study as a pilot and general proof-of-concept to look if the general idea of effects of herbivory on soil carbon storage is possible. We disagree that it is impossible to draw conclusions on the influence of herbivory with our dataset. For the revised version we will make this clearer and discuss the effects of spatial heterogeneity, soil moisture regime, and site history in more detail.

However, due to the consistency found in the differences in both active layer depth and organic carbon content between different grazing intensities across two landscape types, we would argue that herbivory – most likely combined with effects of hydrology – is still a likely explanation.